# GAMap: Zero-Shot Object Goal Navigation with Multi-Scale Geometric-Affordance Guidance

**Shuaihang Yuan**[*1,2,4]**, Hao Huang**[*2,4]**, Yu Hao**[2,3,4]**, Congcong Wen**[2,4]

**Anthony Tzes**[1,2]**, Yi Fang** [†1,2,3,4]

[1]NYUAD Center for Artificial Intelligence and Robotics (CAIR), Abu Dhabi, UAE.
[2]New York University Abu Dhabi, Electrical Engineering, Abu Dhabi 129188, UAE.
[3]New York University, Electrical & Computer Engineering Dept., Brooklyn, NY 11201, USA.
[4]Embodied AI and Robotics (AIR) Lab, NYU Abu Dhabi, UAE.

## Abstract

Zero-Shot Object Goal Navigation (ZS-OGN) enables robots or agents to navigate toward objects of unseen categories without object-specific training. Traditional approaches often leverage categorical semantic information for navigation guidance, which struggles when only objects are partially observed or detailed and functional representations of the environment are lacking. To resolve the above two issues, we propose *Geometric-part and Affordance Maps* (GAMap), a novel method that integrates object parts and affordance attributes as navigation guidance. Our method includes a multi-scale scoring approach to capture geometric-part and affordance attributes of objects at different scales. Comprehensive experiments conducted on HM3D and Gibson benchmark datasets demonstrate improvements in Success Rate and Success weighted by Path Length, underscoring the efficacy of our geometric-part and affordance-guided navigation approach in enhancing robot autonomy and versatility, without any additional object-specific training or fine-tuning with the semantics of unseen objects and/or the locomotions of the robot. Our project is available at `https://shalexyuan.github.io/GAMap/`.

## 1 Introduction

Zero-Shot Object Goal Navigation (ZS-OGN) is a pivotal research domain in embodied AI and robotics, enabling robots to navigate towards the objects of unseen categories without training or fine-tuning on these objects [1, 2, 3, 4, 5]. This capability is crucial for real-world robots, such as home service robots and blind guiding robots, allowing them to interact with diverse objects in real-world scenarios, thereby enhancing their autonomy and versatility.

Prior works on ZS-OGN either leverage deep neural networks to directly map RGB-D observations to actions learned from paired training data [6, 7, 8, 9, 10, 11] or utilize map-based navigation methods [12, 13, 14, 15, 16, 17, 3]. However, deep neural network approaches often struggle due to their dependence on extensive annotated data, resulting in poor generalization to unseen environments [18, 16], while map-based navigation methods instead offer an alternative. Map-based navigation methods track categorical semantics and topological information observed by the agent to select promising exploration locations [17]. With the advent of foundation models, the studies [3, 17, 19] have exploited the reasoning capabilities of Large Language Models (LLMs) to strategically select

---

[*]Equal contribution.
[†]Corresponding author: Yi Fang <yfang@nyu.edu>.

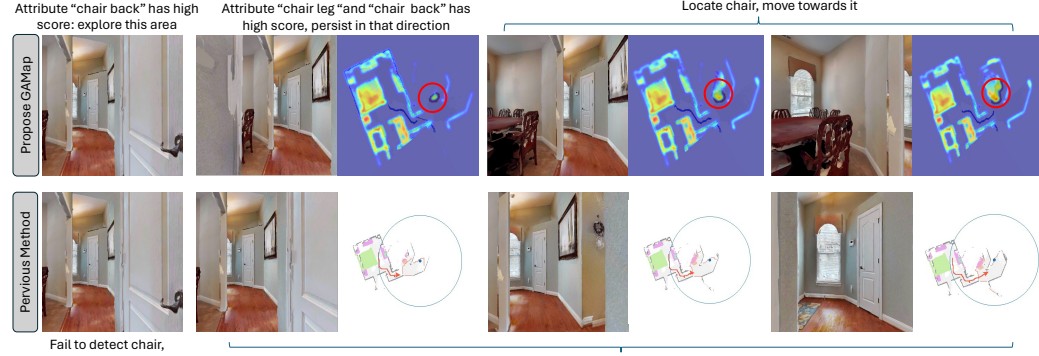

Figure 1: The leftmost RGB image shows the same observation for both methods. Our method (top row) effectively identifies the geometric part of the chair back, which is missed by the traditional method (bottom row). Consequently, GAMap successfully guides the agent to the target object, while the traditional method fails. The red circles highlight the areas where the chair is located, and the GA score is high, indicating the effectiveness of our approach in localizing relevant regions.

waypoints by analyzing commonsense, such as object co-occurrence relationships, to navigate robots towards the target. However, LLM-based approaches require converting visual and semantic information into categorical descriptions, which leads to a loss of spatial and visual information [9]. Vision Language Models (VLMs) enhance semantic reasoning capabilities, but still rely on maps that encompass only categorical information [16]. The primary limitation of exclusively relying on categorical information is that such maps treat objects as *monolithic* entities, disregarding *local* geometric features. This becomes particularly problematic when only the target object is partially observed, leading to incorrect categorical information and potential errors in waypoint selection.

We argue that using a categorical map for robot exploration is suboptimal, as it discards intricate geometric details and functional representations of the environment, as illustrated in Figure 1. Drawing inspiration from human cognitive processes — where distinctive geometric parts are often identified first when locating an object in an unfamiliar environment [20, 21] — we propose *Geometric-part and Affordance Maps* (GAMap), a zero-shot approach, for the geometric parts and affordance attribute driven semantic navigation to explore and find the target object in an unseen environment. Specifically, given a target object, our proposed method starts by using an LLM to infer the object's geometric parts and potential affordance attributes, aiming at providing a detailed understanding of both the object's physical structure and its functional properties. Given depth observations, GAMap maintains a 2D projection of obstacles and explored areas. Instead of relying on object detection and prompt engineering of LLMs to select the next area to explore, our approach employs a pre-trained CLIP [22] to score observations based on their similarity to the reasoned geometric parts and affordance attributes, guiding the exploration process. To construct the proposed GAMap, which requires obtaining scores for geometric parts at different scales, we further propose a Multi-scale Geometric and Affordance score (GA score). Such an integration addresses a notable limitation in existing approaches that compute similarity at a single scale, which usually results in the oversight of fine-grained details of an object, as such details are potentially essential for an accurate identification of geometric parts or affordance attributes of objects with different sizes.

Our proposed method is evaluated on HM3D [23] and Gibson [24] benchmarks and achieves significant improvements in Success Rate (SR) of 26.4% on HM3D [23] and 23.7% on Gibson [24] compared to previous approaches. Additionally, we achieve substantial gains in Success weighted by Path Length (SPL) of 37% on Gibson, highlighting the efficiency and effectiveness of our proposed geometric parts and affordance guided navigation. The contributions of our method are mainly summarized as follows:

1. We propose a novel Geometric-part and Affordance Map (GAMap) for ZS-OGN using object part and affordance attributes as guidance. To the best of our knowledge, this is the first work to study the integration of these attributes in ZS-OGN.

2. Recognizing that geometric parts and affordance attributes often relate to multiple scales of an object, we propose a Multi-scale Geometric and Affordance score, which allows GAMap to be constructed in real-time, better capturing these attributes at different scales.

3. We achieve state-of-the-art performance on two navigation benchmark datasets without any training or fine-tuning with the semantics of unseen objects and/or the locomotions of the robot, which demonstrates the effectiveness of our method in unseen environments.

## 2 Related Work

**Semantic Mapping.** In the context of object goal navigation, it is crucial to transform observations into structured information for effective decision-making. Frontier-based methods [3, 16, 19] utilize categorical semantic information near frontiers to select exploration areas. Additionally, graph-based mapping methods [17, 25, 18] predict waypoints from RGB-D images or simplified maps to create topological representations of the environment. Most of the aforementioned works rely on semantic segmentation or object detection to build semantic maps, which are constrained by the pre-defined semantic classes and thus fail to capture the full semantic richness of environments [18, 26]. To overcome these limitations, recent approaches like VLMaps [18] have introduced open-vocabulary semantic maps, enabling natural language indexing and expanding the scope of semantic mapping. In addition, previous works [27, 28] attempt to utilize attributes and long descriptions for object perception. While these methods have advanced the navigation field, they often overlook object parts, treating objects as monolithic entities and leading to errors when these objects are partially observed. Inspired by human cognitive processes [21, 20], where distinctive geometric parts are identified first in unseen environments, we propose Geometric-part and Affordance Maps (GAMap). Unlike previous methods focusing solely on categorical information, GAMap integrates geometric parts and affordance attributes, providing a richer and more functional representation of the environment.

**Zero-shot Object Goal Navigation.** In the context of object goal navigation, the aim is to efficiently explore a new environment while searching for a target object that is not directly visible. Previous research relies heavily on visual context via imitation [29, 6] or reinforcement learning [7, 8, 9] to guide robots. These approaches often require extensive data collection and annotation for training, which limits their practical application in real-world environments. Thus, the focus in object goal navigation has been shifting towards zero-shot object navigation, which aims to equip robots with the ability to adapt to unseen objects and environments without the need for training [12, 30, 13, 14]. Clip-Nav [31] utilizes CLIP [22] to execute vision-and-language navigation in a zero-shot scheme, whilst CoW [32] employs CLIP for object goal navigation. Recently, Frontier-based Exploration (FbE) [33] is widely adopted in navigation by moving the robot to the frontier between known and unknown spaces [32, 34, 35, 36, 37], leading to promising performance compared to learning-based exploration methods [38, 39]. More recently, ESC [3] leverages the reasoning ability of LLMs to select frontiers using pre-defined rules. Chen *et al*. [15] explore frontier selection by jointly considering the shortest path to frontiers and the relevance scoring between objects for exploration. To enable more robust and reliable exploration and waypoint selection, Wu *et al*. [17] propose a Voronoi-based scene graph for waypoint selection. Unlike the above methods that use the reasoning ability of LLMs to select frontiers, VLFM [16] introduces a value map to score frontiers based on the categorical similarity between the observation and the target object. In contrast to prior work, for the first time, we explore robot navigation using geometric parts and affordance attributes as guidance. This approach integrates detailed geometric parts and functional properties of objects, offering a more comprehensive strategy for navigation.

## 3 Method

We first formalize the ZS-OGN problem in Section 3.1. Then, we detail our method, as shown in Figure 2, from four phases: attribute generation in Section 3.2, multi-scale attribute scoring in Section 3.3, GAMap generation in Section 3.4, and exploration policy in Section 3.5. Initially, our method generates geometric parts and affordance attributes for the target object. During the exploration, the method computes a multi-scale attribute score from the RGB observations collected by the agent. These scores are then mapped onto a 2D geometric parts and affordance map, which is pivotal in guiding the exploration process. Subsequently, the agent selects the location with the highest score for further exploration.

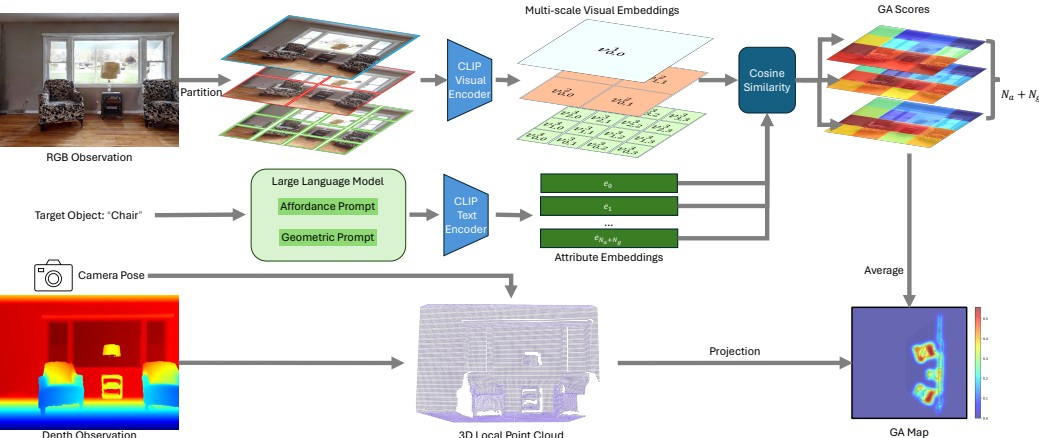

Figure 2: Pipeline of the GAMap generation. Geometric parts and affordance attributes are generated by an LLM. The RGB observation is partitioned into multiple scales, and a CLIP visual encoder generates multi-scale visual embeddings. GA scores are computed using cosine similarity between attribute text embeddings from a CLIP text encoder and the multi-scale visual embeddings. These scores are averaged and projected onto a 2D grid to form the GAMap.

## 3.1 Problem Formulation

In ZS-OGN, the robot must navigate to a target object $g_i$ that has never been encountered before in an unseen environment $s_i$, without any training on $g_i$ and $s_i$. A navigation episode can be defined as $\mathcal{E}_i = \{g_i, s_i, p_0\}$, where $p_0$ denotes the robot's initial location, the subscript $i$ refer to the $i^{th}$ episode. The robot receives a color image $I_t$, a depth image $D_t$, and its pose, *i.e.*, position $(x_t, y_t)$ and orientation $\theta_t$, at each exploration step $t$. We denote these readings as an observation $\mathcal{O}_t = \{I_t, D_t, x_t, y_t, \theta_t\}$. The agent accumulates pose readings over time to determine its relative position $p_t$. Based on the readings at each step, the robot needs to select an action $a_t$ from the action space. A navigation episode is marked as successful if the robot executes the STOP action within a pre-defined distance to the target object. In this work, we approach the navigation task as a sequence of decisions made by the robot. The process starts at the initial time step $t = 0$ and ends at the final time step $T$. This final step is either when the robot executes the STOP action or when a pre-defined maximum number of exploration steps is reached.

## 3.2 Attribute Generation

We focus on two types of attributes essential for object recognition: *Affordance* and *Geometric-part* attributes. Affordance attributes refer to the potential actions that an object facilitates [40], which are crucial for understanding how an agent might interact with different objects within an environment. Geometric-part attributes, on the other hand, describe the shape and spatial characteristics of an object, aiding in its visual identification and differentiation from other objects.

To extract these attributes, we employ an LLM to reason about the target object's characteristics. Specifically, we utilize GPT-4V [41] for the attributes generation. We initiate this process by setting the system prompt as: *"I am a highly intelligent question-answering bot, and I answer questions from a human perspective."* Subsequently, we employ two tailored prompts to extract the desired attributes. For affordance attributes, we design prompt as: *"For the target object <target object $g_i$>, please provide <$N_a$> affordance attributes that to the most reflect its characteristics."* to query $N_a$ number of affordance attributes. For geometric parts, the prompt is: *"Summarize <$N_g$> geometric part visual features of the <target object $g_i$> which are typically used to identify why it is a <target object $g_i$>."* to get $N_g$ number of geometric attributes. Once the set of affordances attributes $\{A_n\}_{n=1}^{N_a}$ and geometric attributes $\{G_n\}_{n=1}^{N_g}$ have been identified, the next stage involves coupling these attributes with the agent's observations by computing Geometric-part and Affordance scores (GA scores) through multi-scale visual features, as detailed in the following.

## 3.3 Multi-scale Attribute Scoring

The key idea of Multi-scale Attribute Scoring is to quantify the relevance of observed areas' geometric and affordance characteristics in locating the target object. These scores determine the agent's subsequent exploration decisions, guiding it to the areas most likely to contain the target object.

To correlate the scoring with both the global frame and localized patches, ensuring that finer details and smaller components are effectively scored, we partition the observed image into patches of equal size across different scales. Specifically, given an RGB observation image $I_t$, with height $H$ and width $W$, the partitioning process is as follows. At level $k$, the image is partitioned into $4^{(k-1)}$ equal parts with each patch at level $k$ of size $\frac{H}{2^{(k-1)}} \times \frac{W}{2^{(k-1)}}$. The patches at level $k$ can be represented as: $I_t^k = \left\{ P_{h,w} \mid 1 \leq h \leq 2^{(k-1)}, 1 \leq w \leq 2^{(k-1)} \right\}$ where $P_{h,w}$ denotes the patch located at the $h^{th}$ row and $w^{th}$ column of the partitioned image at level $k$. All patches from all levels are then resized to the same dimensions for further processing. We utilize CLIP [32] to calculate the visual embeddings for all patches from all levels. For each patch $P_{h,w}^k$ at level $k$, its visual embedding $v_{h,w}^k$ is computed using the image encoder of CLIP. Simultaneously, each attribute embedding $e$ is computed using the text encoder of CLIP, given the attribute descriptions, $\{A_n\}_{n=1}^{N_a}$ and $\{G_n\}_{n=1}^{N_g}$, generated from an LLM. For each patch $P_{h,w}^k$, we calculate the cosine similarity between its visual embedding $v_{h,w}^k$ and the attribute embedding $e$ to obtain a similarity score $S_{h,w}^{k,e}$ as follows:

$$S_{h,w}^{k,e} = \frac{v_{h,w}^k \cdot e}{\|v_{h,w}^k\| \|e\|} \quad . \tag{1}$$

Due to the hierarchical partitioning of the image, we accumulate the scores across all scales for each pixel location and then take the average to obtain the final score for the image. Specifically, let $L$ be the number of levels, and the accumulated score for the pixel at position $(p, q)$ in the image $I_t$ is calculated by summing the scores from all levels and then averaging:

$$S(p, q, e) = \frac{1}{L} \sum_{k=1}^{L} S_{h(p,q,k),w(p,q,k)}^{k,e} \quad , \tag{2}$$

where $h(p, q, k)$ and $w(p, q, k)$ map the pixel position $(p, q)$ in the image to the corresponding patch indices at level $k$. The scores are then used to generate a Geographic and Affordance Map (GAMap) for the explored area. The process for generating the GAMap is detailed in the next section.

## 3.4 Geographic and Affordance Map

At the core of our approach is the Geographic and Affordance Map. This map assigns a GA score to each pixel within the explored area to quantify the relevance of different regions in locating the target object, associating the areas with the highest values as the most promising for further exploration.

We define the GAMap at time step $t$ as $M_t \in \mathbb{R}^{\hat{H} \times \hat{W} \times C}$, where $\hat{H}$ and $\hat{W}$ are the dimensions of the 2D projection grid map, and $C$ is the number of attributes, $i.e.$, $C = N_a + N_g$ with $N_a$ and $N_g$ representing the number of attributes and parts, respectively. To construct the GAMap from the RGB-D observation $I_t$ and the depth data $D_t$, we back-project every pixel from $D_t$ to reconstruct the point cloud following [18]:

$$\mathbf{X} = D_t(p, q) \cdot \mathbf{K}^{-1} \cdot [p, q, 1]^T \quad , \tag{3}$$

where $\mathbf{K}$ is the intrinsic matrix of the depth camera, and $D_t(p, q)$ is the depth value of the pixel at the coordinate $(p, q)$. To transform the point cloud into the world coordinate frame, we use $\mathbf{X_{world}} = \mathbf{T_W} \cdot \mathbf{X}$, where $\mathbf{T_W}$ is the transformation matrix from the camera coordinate to the world coordinate. The 3D points are then projected onto the ground plane to determine the corresponding positions on the grid map. We assume perfect calibration between the depth and RGB cameras, allowing us to project each image pixel's score to its corresponding grid cell in the map. Given that multiple 3D points may project to the same grid location, we retain the maximum value for each channel that falls into the same grid cell as the score of this cell:

$$M_t(\hat{h}, \hat{w}, e) = \max \left\{ S(p, q, e) \mid (p, q) \in \text{cell } M_t(\hat{h}, \hat{w}) \right\} \quad , \tag{4}$$

where $M_t(\hat{h}, \hat{w}, e)$ represent the score of attribute $e$ in the 2D grid at position $(\hat{h}, \hat{w})$, $S(p, q, e)$ is the score for the pixel at $(p, q)$ for the attribute $e$, and cell $M_t(\hat{h}, \hat{w})$ denotes the grid cell on the GAMap. When the robot moves to a new position, resulting in overlapping observations with the previously explored regions, the GA scores for each pixel in the overlapping region are updated. The updated GAMap is computed by taking the maximum of the current attribute score and the previous attribute score for each cell:

$$M(\hat{h}_o, \hat{w}_o, e) = \max\left(M_t(\hat{h}_o, \hat{w}_o, e), M(\hat{h}_o, \hat{w}_o, e)\right) \quad , \tag{5}$$

where $M(\hat{h}_o, \hat{w}_o, e)$ is the GAMap constructed from the previous step, and the subscript $o$ represents the overlapped grid cell.

## 3.5 Exploration Policy

To determine the next area to explore, the robot selects the region with the highest GA score, calculated as the average of all attribute channels. To enable efficient exploration, only areas near the frontier with the highest scores are selected. Once the area with the highest GA score is identified, a heuristic search algorithm, *i.e.*, the Fast Marching Method (FMM) [42], is employed to find the shortest path from the robot's current location to the selected area. The robot then generates the appropriate actions to navigate along this path. At each step, the GAMap is updated based on new observations. We repeat this process until the robot either identifies and reaches the target object or the episode ends.

## 4 Experiment

**Datasets. HM3D** [23] is a dataset consisting of 3D data from real-world indoor spaces, along with semantic annotations, serving as a foundational resource for the Habitat 2022 ObjectNav Challenge [23]. This comprehensive dataset includes 142,646 object instance annotations, organized into 40 distinct classes across 216 environments, covering a total of 3,100 individual rooms. We follow the validation settings from [3, 32] to evaluate our proposed method. **Gibson** [24] was developed by Al-Halah *et al.* [43]. The dataset comprises 5 validation scenes across 6 object categories, and we adhere to the standard evaluation protocol [44, 19, 30, 2] to use all 5 validation scenes for evaluation.

**Metrics. Success Rate (SR, %)** [45] focuses on the agent's accuracy in reaching the designated target, where a higher value indicates better performance. SR is computed based on whether the robot successfully stops within 0.1m of the target object $g_i$:

$$SR(\pi) = \frac{1}{K} \sum_{i=1}^{K} \mathbf{1}_{\{d(g_i, p_{T_i}) \leq 0.1\}} \quad , \tag{6}$$

where $K$ is the number of episodes, $d(g_i, p_{T_i})$ is the distance between the target object $g_i$ and the robot's final position $p_{T_i}$ in the $i^{th}$ episode, and $\mathbf{1}_{\{.\}}$ is an indicator function. **Success weighted by Path Length (SPL, %)** [45] evaluates success relative to the shortest possible path, normalized by the actual path taken by the agent, measuring the efficiency of the agent's success in reaching a goal, defined as:

$$SPL(\pi) = \frac{1}{K} \sum_{i=1}^{K} \mathbf{1}_{\{d(g_i, p_{T_i}) \leq 0.1\}} \cdot \frac{L_i^*}{\max(L_i, L_i^*)} \quad , \tag{7}$$

where $L_i$ is the actual path length traveled by the robot in the $i^{th}$ episode and $L_i^*$ is the shortest possible path length to the target in the same environment.

## 4.1 Baselines

We compare our method against several ZS-OGN approaches, including the state-of-the-art methods: **Random Exploration**: takes random actions to explore the environment. **Nearest FbE** [33]: explores the environment by selecting the nearest frontier. **SemExp** [2]: utilizes a category semantic map and trains a local navigation policy for exploration. **PixNav** [9]: trains models for navigation by selecting pixels as intermediate goals. **PONI** [44]: uses potential functions to select frontiers for exploration. **ZSON** [30]: employs categorical information to train a model for object-based navigation tasks. **CoW**

[32]: uses CLIP for categorical information extraction and explores using the nearest frontier-based exploration. **ESC** [3]: utilizes a categorical semantic map and commonsense reasoning for target object exploration. **L3MVN** [19]: uses an LLM to reason about the next exploration area based on a trained detection head for semantic map construction. **VLFM** [16]: employs BLIP-2 [46] and categorical information of the target object to evaluate and select frontiers for exploration. **VoroNav** [17]: uses a Voronoi-based decomposition strategy for navigation. **SemUtil** [15]: considers the shortest path distance to frontiers and the relevance scoring between objects for exploration.

Table 1: Comparison of navigation performance between different methods on HM3D and Gibson datasets, measured by SR and SPL metrics. This table highlights the performance of our proposed method, demonstrating improvements over existing methods on both datasets.

| Method | Venue | Zero-shot | Training | | HM3D | | Gibson | |
| --- | --- | --- | --- | --- | --- | --- | --- | --- |
| | | | Locomotion | Semantic | SR↑ | SPL↑ | SR↑ | SPL↑ |
| SemExp [2] | NeurIPS 20 | ✗ | ✓ | ✓ | 37.9 | 18.8 | 65.2 | 33.6 |
| ZSON [30] | NeurIPS 22 | ✗ | ✓ | ✓ | 25.5 | 12.6 | 31.3 | 12.0 |
| PixNav [9] | ICRA 24 | ✗ | ✓ | ✗ | 37.9 | 20.5 | - | - |
| VLFM [16] | CoRL 23 | ✓ | ✓ | ✗ | 52.5 | **30.4** | *84.0* | *52.2* |
| PONI [44] | CVPR 22 | ✗ | ✗ | ✓ | - | - | 73.6 | 41.0 |
| FbE | - | ✓ | ✗ | ✓ | 23.7 | 12.3 | 41.7 | 21.4 |
| L3MVN [19] | IROS 23 | ✓ | ✗ | ✓ | 50.4 | 23.1 | 76.1 | 37.7 |
| Random | - | ✓ | ✗ | ✗ | 0.0 | 0.0 | 3.0 | 3.0 |
| CoW [32] | CVPR 23 | ✓ | ✗ | ✗ | 32.0 | 18.1 | - | - |
| ESC [3] | ICML 23 | ✓ | ✗ | ✗ | 38.5 | 22.0 | - | - |
| SemUtil [15] | RSS 23 | ✓ | ✗ | ✗ | - | - | 69.3 | 40.5 |
| VoroNav [17] | ICML 24 | ✓ | ✗ | ✗ | 42.0 | 26.0 | - | - |
| VLFM Value Map + FMM [16] | CoRL 23 | ✓ | ✗ | ✗ | *50.9* | 23.6 | *82.8* | *48.5* |
| **GAMap** | Proposed | ✓ | ✗ | ✗ | **53.1** | *26.0* | **85.7** | **55.5** |

## 4.2 Results and Analysis

We compare our method with existing approaches across four categories: those utilizing both locomotion and semantic training [2, 30], those employing only locomotion training [9, 16], those using only semantic training [19, 44], and those that do not incorporate any training [3, 32, 17, 15]. Locomotion training involves learning-based methods for navigation, while semantic training requires training or fine-tuning a perception module to construct a semantic map. The results and comparisons are shown in Table 1.

On the HM3D dataset, our method achieved a SR of 53.1% and a SPL of 26.0%. This represents a significant improvement over the best method [17] that does not use locomotion and semantic training, with a 26.4% increase in SR. Although the SPL (26.0%) of our method is lower than that of VLFM (30.4%), this discrepancy can be attributed to the fact that VLFM's local policy planning is trained. Considering the different path planning methods adopted in our approach and VLFM, we construct a com-

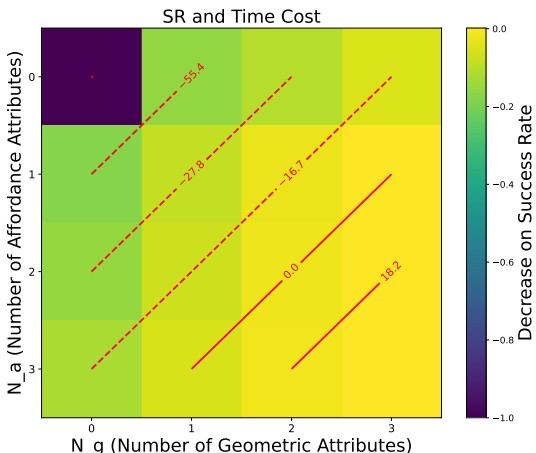

Figure 3: Heatmap showing the increase and decrease in the percentage of SR and time cost for varying the numbers of $N_a$ and $N_g$. Darker colors indicate a greater decrease in SR, and red solid and dashed lines represent the associated time cost.

parative experiment with VLFM. Specifically, we kept the VLFM value map generation process unchanged and replaced its path planning method with FMM instead of the trained policy. The detailed results are shown in the second last row in Table 1. The comparison reveals that our model outperforms VLFM-based mapping method by 4.32% and 10.17% in SR and SPL on HM3D. On the Gibson dataset, GAMap attained an SR of 85.7% and an SPL of 55.5%, which marks a substantial

improvement over methods that do not utilize locomotion and semantic training [17], with a 23.7% increase in SR and a 37.0% increase in SPL.

# 5 Ablative Study

## 5.1 Effectiveness of Affordance and Geometric-part Attributes

We analyzed the effectiveness of the number of affordances and geometric parts in our proposed method. Ablation studies were conducted by varying $N_a$ and $N_g$ to assess performance gain versus time loss, as well as the contribution of each attribute type. We evaluated on the official mini-validation split of the HM3D dataset.

The heatmap, as shown in Figure 3, illustrates the SR with different combinations of $N_a$ and $N_g$. The color changes from dark purple to yellow indicate the increased percentage in SR. Red sold and dashed lines with labels indicate the time cost associated with each combination of $N_a$ and $N_g$. Increasing the number of geometric parts ($N_g$) from 0 to 3 results in a significant improvement in SR across all levels of $N_a$. For example, when $N_a = 0$, increasing $N_g$ from 0 to 3 raises the SR more than increasing $N_a$ from 0 to 3 when $N_g = 0$. This demonstrates the substantial contribution of geometric parts to navigation performance. To more clearly demonstrate the specific impacts of $N_a$ and $N_g$ on various performance metrics, we have converted Figure 3 to Table 6 in the Appendix. Similarly, increasing the number of affordance attributes ($N_a$) also improves SR, though the effect is slightly less pronounced than that of geometric parts. The best performance is achieved when both $N_a$ and $N_g$ are maximized, with both set to 3. This suggests a synergistic effect, where the combination of both attributes leads to optimal navigation performance. However, it requires an 18.2% increase in time cost.

## 5.2 Effectiveness of Different Scaling Levels

We analyzed the effectiveness of different scaling levels ($L$). Figure 4 presents changes in SR and SPL when using different numbers of scales and the time required for processing on the mini-validation split of HM3D.

Increasing the number of scaling levels from 1 to 4 leads to notable changes in both SR and SPL. At the highest scale level of 4, the SR improves by approximately 10%, and the SPL increases by about 20% than $L = 1$. However, there is a trade-off between performance improvement and time cost. The time required increases with the number of scales, as indicated by the red line in the figure. The time cost starts at 0.66 seconds for a single scale and rises progressively, reaching 0.78 seconds at the fourth scale level. This increase in time cost suggests that while higher scaling levels improve both SR and SPL, they also

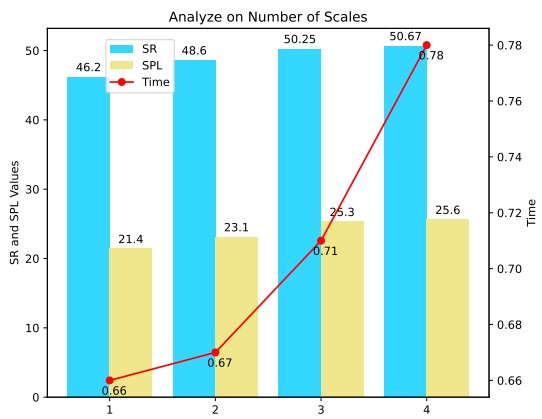

Figure 4: Changes in SR, SPL, and processing time across different scaling levels on the mini-validation split of HM3D. Increasing scales improves SR and SPL but also increases processing time.

demand more computational resources and time. This emphasizes the need to balance performance and computational efficiency when determining the optimal scaling level for practical applications.

## 5.3 Effectiveness of Different Methods for Calculating GA Scores

We analyzed the proposed patch-based method by comparing it with the gradient-based method. The visualization of the GA score for different methods is shown in Figure 5, which illustrates the GA score of the armrest, backrest, and seat attributes of a target object chair. Observations indicate that the gradient-based method often attends to irrelevant areas. For example, the ceiling of the room has a higher GA score, which is incorrect. In contrast, the patch-based method more accurately focuses

on relevant areas, such as the armrest, backrest, and seat of the chair, validating its effectiveness over the gradient-based method. Moreover, as observed from Table 2, the gradient-based method is also slower than the patch-based method. One reason for this is that the gradient-based method requires back-propagation of the gradient.

Based on different methods, we further explored the effectiveness of various pre-trained encoders, as shown in Table 2. We compared SR and computation time required by three types of encoders: CLIP (ours), BLIP, and BLIP-2. Although using more powerful encoders such as BLIP and BLIP-2 leads to better performance, they require significantly more time than CLIP, while the performance gain is limited. This trade-off makes it less valuable for us to use a BLIP-based encoder.

Moreover, we analyzed different ways of aggregating GA scores using patch-based methods. Given the multiple levels of patches, we aimed to find the best method to aggregate the final GA score from multi-level GA scores.

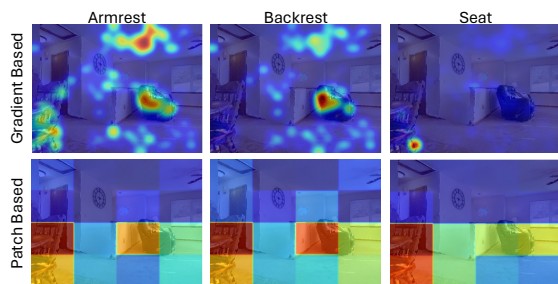

Figure 5: Comparison of GA score visualization between gradient-based and patch-based methods for the armrest, backrest, and seat attributes of a target chair. The gradient-based method (top row) often attends to irrelevant areas, such as the ceiling, while the patch-based method (bottom row) accurately focuses on the relevant areas.

We compared "Max Value" and "Average Value" (ours). In the "Max Value" method, the maximum GA score across different scale levels is used. The "Average Value" method, which we propose, calculates the average GA score across levels. Note that the gradient-based method directly gives the score, so it is not analyzed. As shown in Table 2, using the average to aggregate the score gives us the best performance.

Table 2: Analysis of different GA scoring methods.

| Method | Encoder | Ave.↑ | Max↑ | Time↓ |
|---|---|---|---|---|
| **Patch** | CLIP | 50.25 | 49.75 | 0.7 |
| | BLIP | 51.20 (↑≈1%) | 50.67(↑≈1%) | 1.2 (↓71%) |
| | BLIP-2 | 51.60 (↑≈2%) | 51.10 (↑≈2%) | 1.6 (↓128%) |
| **Gradient** | CLIP | 49.78 (↑<1%) | | 0.9 (↓71%) |
| | BLIP | 50.67 (↑≈1%) | | 1.5 (↓114%) |

Table 3: Impact of different GA score updating methods.

| Method | SR | SPL |
|---|---|---|
| Max | 50.25 | 0.253 |
| Average | 48.30 (↓≈3%) | 0.226 (↓≈10%) |
| Replacement | 47.78 (↓≈4%) | 0.209 (↓≈17%) |

## 5.4 Effectiveness of Different Methods for Updating GA Scores

Different methods of updating GA scores and their impact on navigation performance are shown in Table 3. In the "Replacement" method, the previous value is disregarded and overwritten with the new one. The "Average" method calculates the new value as the average of the previous and current values. Our approach, "Max", retains the maximum scores between the previous and new values, which memorizes the most salient score in a specific direction, as the agent could observe an object from different perspectives during exploration, thus potentially finding the optimal perspective.

Our findings indicate that the "Max" method consistently enhances performance compared to the other two methods across all three datasets. In Table 3, the "Max" method achieves SR of 50.25% and SPL of 25.3%. The "Average" method results in a 3% decrease in SR and a 10% decrease in SPL, indicating a moderate impact on performance. The "Replacement" method shows the most significant performance drop, with a 4% decrease in SR and a 17% decrease in SPL. These results highlight that the "Max" method is the most effective in maintaining and enhancing navigation performance, as it better captures and retains the most relevant object attributes from different perspectives.

## 5.5 Effectiveness of Geometric and Affordance Guidance Navigation

We evaluated the effectiveness of the proposed part and affordance guidance navigation by analyzing three types of errors: detection error, planning error, and exploration error. 1) *Detection error* happens

when the agent either misses the goal or incorrectly believes it has detected the goal. 2) *Planning error* arises when the agent either recognizes the target but cannot reach it or gets stuck without spotting the goal, reflecting the path-planning ability of the system. 3) *Exploration error* occurs when the agent fails to see the goal object due to issues other than planning or detection, assessing its ability to approach the goal. Table 4 shows the comparison of these errors using our proposed method versus using categorical semantic information as the guidance for navigation. Note that our proposed method significantly decreases the errors in all three categories.

## 5.6 Effectiveness of Multi-scale Approach

Furthermore, we conduct an experiment to verify whether VLM [47] models can directly capture enough multi-scale information. We randomly selected a scene and compared the ability of GPT-4V and our

Table 4: Comparison of errors using categorical semantic guidance versus geometric parts and affordance guidance.

| Method | Detection Error (%) | Planning Error (%) | Exploration Error (%) |
|---|---|---|---|
| Categorical | 16.95 | 19.1 | 15.65 |
| GA | 9.75 (↓42.5%) | 11.34 (↓40.6%) | 12.78 (↓18.3%) |

proposed CLIP with multi-scale scoring method to identify the target object. As shown in Figure 6, we input an image with a sofa located in a distant corner as the target object and compared the subsequent movement trajectories of the two algorithms. As illustrated in Figure 6, our method successfully captures the small sofa back in the far corner, leveraging geometric parts and affordance attributes to guide the exploration process. In contrast, GPT-4V failed to identify the object.

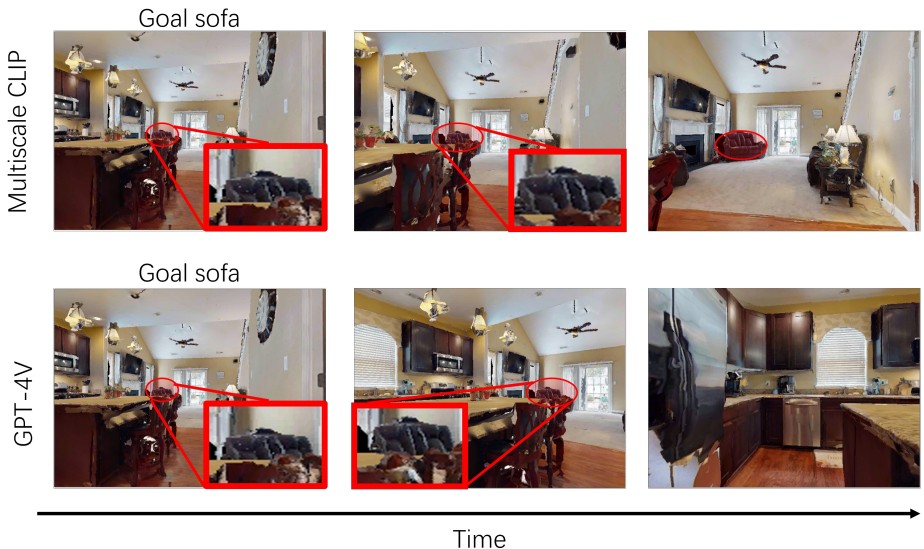

Figure 6: The top row of images shows our proposed method, where the multi-scale approach effectively captures objects at all scales, such as the sofa back in the background. The bottom row of images shows the results of GPT-4V.

## 6 Conclusion

In this work, we introduced the Geometric-part and Affordance Maps (GAMap) for zero-shot object goal navigation, leveraging geometric parts and affordance attributes to guide exploration in unseen environments. Our method employs LLMs for detailed attribute inference and VLMs for multi-scale scoring, capturing object intricacies at various scales. Comprehensive experiments on HM3D and Gibson datasets exhibit significant improvements in SR and SPL over previous methods. These results highlight the effectiveness of our approach in enhancing navigation efficiency without any task-specific training or fine-tuning.

## Acknowledgements

Authors appreciate the support provided by the NYUAD Center for Artificial Intelligence and Robotics (CAIR), funded by Tamkeen under the NYUAD Research Institute Award CG010.

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

## A   Difference with Existing Works

As shown in Table 5, the key difference between our method and other methods is that we leverage geometric parts and affordance information to represent the environment, in addition to using object-level category information as previous methods do. Furthermore, we utilize multi-scale feature representation to capture local features, enhancing the overall accuracy and robustness.

Table 5: The differences between our work and existing methods.

| Method | Mapping | Multi-Scale | Zero-shot | Training | |
|---|---|---|---|---|---|
| | | | | Locomotion | Semantic |
| SemExp [2] | Categorical | × | × | ✓ | ✓ |
| ZSON [30] | Categorical | × | × | ✓ | ✓ |
| PixNav [9] | Language-Grounded | × | ✓ | ✓ | × |
| VLFM [16] | Language-Grounded | × | ✓ | ✓ | × |
| PONI [44] | Categorical | × | ✓ | × | ✓ |
| L3MVN [19] | Categorical | × | ✓ | × | ✓ |
| CoW [32] | Categorical | × | ✓ | × | × |
| ESC [3] | Categorical | × | ✓ | × | × |
| VoroNav [17] | Categorical | × | ✓ | × | × |
| **GAMap** | Affordance+Gemoetric | ✓ | ✓ | × | × |

## B   Experiment Setup

In our experiment, we adopt GPT-4V as the LLM to generate the geometric and affordance attributes. We set $N_a$ to 1 and $N_g$ to 3 for the experiments on the HM3D and Gibson datasets. For the partition process, we use three scaling levels in all our experiments: the first level is the original image, the second level has 4 equal-sized patches, and the third level has 16 equal-sized patches. We use CLIP as the pre-trained visual and text encoder. Following the standard evaluation protocol [19], we use 2000 episodes on the validation split of HM3D to report the results. Similarly, we follow this method [19]to produce the results on the Gibson dataset. We use a Titan XP GPU for the experiment evaluation, and the entire evaluation process takes around 44 hours.

## C   Effectiveness of Affordance and Geometric-part Attributes

We provide the quantitative result for Figure 3 in the main paper. Time is measured in seconds.

Table 6: Quantitative results for the effectiveness of affordance and geometric-part attributes.

| $N_a$ | $N_g$ | SR | Time | $N_a$ | $N_g$ | SR | Time |
|---|---|---|---|---|---|---|---|
| 0 | 0 | - | - | 2 | 0 | 42.6 | 0.152 |
| 0 | 1 | 42.1 | 0.094 | 2 | 1 | 46.4 | 0.175 |
| 0 | 2 | 44.7 | 0.152 | 2 | 2 | 49.1 | 0.210 |
| 0 | 3 | 47.5 | 0.175 | 2 | 3 | 50.3 | 0.248 |
| 1 | 0 | 41.2 | 0.094 | 3 | 0 | 44.0 | 0.175 |
| 1 | 1 | 45.7 | 0.152 | 3 | 1 | 47.2 | 0.210 |
| 1 | 1 | 48.8 | 0.175 | 3 | 2 | 49.2 | 0.248 |
| 1 | 3 | 50.2 | 0.210 | 3 | 3 | 50.3 | 0.290 |

## D   Result Visualizations

In this section, we visualize the navigation paths on both the Gibson and HM3D datasets, as shown in Figures 7 and 8. Part of the visualization code is adapted from L3MVN [19].

# E   Time Complexity

To validate the efficiency of our method, we compare the FPS of our method to SemExp [2], L3MVN [16], and VLFM [19] in navigation tasks on the HP3D dataset. The experimental results are shown in Table 7. It can be observed that SemExp has the highest FPS, indicating the fastest processing speed. This is because it uses a detection head and does not employ a foundation model. However, SemExp has the lowest SR and SPL, indicating that despite its fast processing speed, it performs poorly in navigation accuracy and path efficiency. In contrast, L3MVN has the second-highest FPS as it uses a lightweight foundation model. Although its processing speed is not as fast as SemExp, it shows improved navigation accuracy and path efficiency, achieving an SR of 76.1% and an SPL of 37.7%. On the other hand, VLFM has a lower FPS of only 2, but it significantly improves SR and SPL, reaching 84.0% and 52.2%, respectively. This indicates that although VLFM has a slower processing speed, it has considerable advantages in navigation accuracy and path efficiency. Our model has the same FPS as VLFM, both at 2, but further improves SR and SPL, reaching 85.7% and 55.5%, respectively. This demonstrates that our method maintains high navigation accuracy and path efficiency while providing comparable processing speed to VLFM. These experimental results verify that our proposed method achieves a good balance between time and accuracy.

Table 7: Comparison of different method's FPS on the HM3D dataset.

| Method | FPS | HM3D | |
| --- | --- | --- | --- |
| | | SR↑ | SPL↑ |
| SemExp [2] | 4 | 37.9 | 18.8 |
| VLFM [16] | 3 | 52.5 | 30.4 |
| L3MVN [19] | 2 | 50.4 | 23.1 |
| GAMap | 2 | 53.1 | 26.0 |

# F   Real-world Experiments

In our real-world experiment, we will evaluate the performance of four zero-shot object goal navigation algorithms, including L3MVN [19], COW [32], ESC [3], and VLFM [16], within a standard indoor apartment environment consisting of two bedrooms, two bathrooms, one kitchen, and one living room. The experiment adopts a four-wheeled robot. Specifically, we use a JetAuto-Pro from Hiwonder equipped with an Intel Realsense D435i camera as our robot agent to navigate the environment and locate specific target objects, including a bed, toilet, table, sofa, and chair, without any prior knowledge of their locations. To ensure a fair comparison, the starting position of the robots was kept consistent across all trials for each algorithm. The video demo can be found on our project page: https://shalexyuan.github.io/GAMap/.

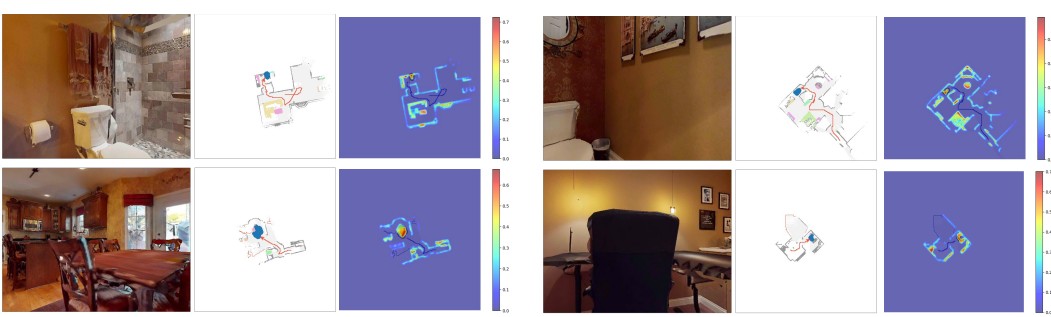

Figure 7: Visualizations of the last observation frame, navigation path, and GAMap.

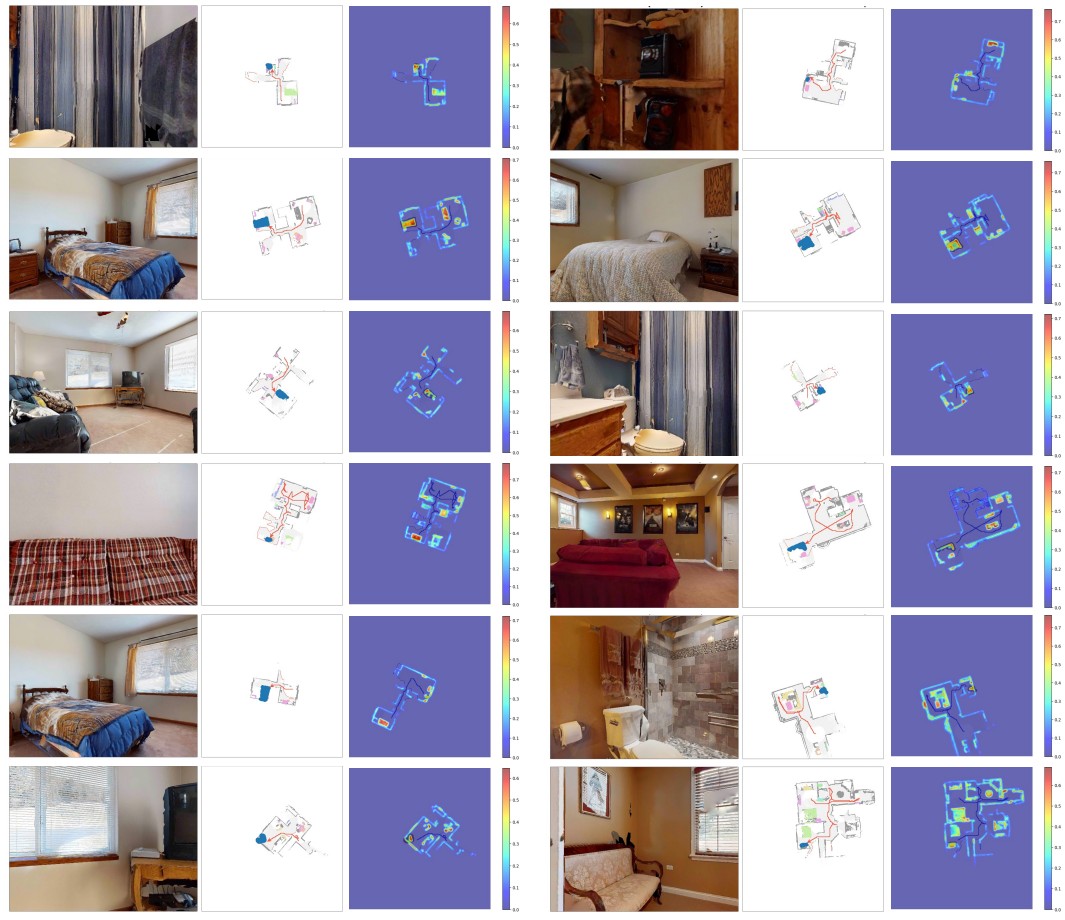

Figure 8: Visualizations of last observation frame, navigation path, and GAMap.

## G   Limitation and Future Work

While our proposed method, Geometric-Affordance Maps (GAMap), has demonstrated significant improvements in zero-shot object goal navigation, it is important to acknowledge its limitations. Our approach relies heavily on the visual processing power of vision-language models, *i.e.*, CLIP. The effectiveness of GAMap also depends on the accuracy of geometric parts and affordance attributes inferred by LLMs. Although the multi-scale scoring method enhances attribute detection, it introduces additional complexity and computational overhead.

To address these limitations and further advance this research field, future work should optimize the integration of LLMs and VLMs to reduce computational overhead, potentially through techniques like model distillation. Enhancing the accuracy of geometric and affordance attribute inference is crucial, and more powerful foundational models could improve this accuracy. Additionally, exploring better methods for aggregating these attributes is also an interesting research direction.

