# OpenReview forum: "GAMap: Zero-Shot Object Goal Navigation with Multi-Scale Geometric-Affordance Guidance"
_NeurIPS.cc/2024/Conference — NeurIPS 2024 poster_

### Official Review · Reviewer_fGDQ · 2024-07-12

**Soundness:** 2
**Presentation:** 2
**Contribution:** 2
**Rating:** 4
**Confidence:** 3

**Summary:**

The paper is studying the problem of zero-shot object goal navigation. To solve the problems in traditional approaches, the authors propose a novel method, named Geometric and Affordance Maps (GAMap), which incorporates both geometric and affordance attributes as navigation guidance. A multi-scale scoring approach is proposed to cope with objects at different scales. Experiments on HM3D and Gibson datasets show the effectiveness of the proposed method for the metrics of success rates and success weighted by path length.

**Strengths:**

1. The paper is studying the problem of zero-shot object goal navigation, which I think is an important problem in object-goal navigation.

2. The proposed method incorporating both geometric and affordance attributes is sensible to me.

3. There are extensive analysis in the experiment section of the paper, which provides several insights.

**Weaknesses:**

1. Although the authors have demonstrated the difference of their work compared with previous ones, I am still not convinced about the key novelty of the work. In terms of organisation of the paper, it would be better if the authors could provide a table to better compare the difference of their work with existing works.

2. There seems no real world experiments. I am not fully convinced about the effectiveness of the proposed method unless results of real robot experiments are provided.

3. The authors have not removed the checklist instruction blocks.

4. In Line 214, there is an issue with the citations. The paper needs further proofread.

**Questions:**

It would be good if the authors could address my concerns in the weakness part. I will make my final decision after reading the authors' rebuttal and other reviewers' comments.

**Limitations:**

The authors have discussed about limitation in appendix.

---

> ### Author Rebuttal · Authors · 2024-08-06
>
> # Response to Weakness 1
>
> Thank you to the reviewer for the thorough review and feedback on our work. First, we would like to clarify the two key innovations of our work:
>
> Our main contribution is the proposed **Geometric and Affordance Map (GAMap)** and the **Multi-scale Geometric and Affordance Scoring** to address the challenges in semantic scene understanding during open-vocabulary robotic exploration in unknown environments, caused by occlusion or partial viewing angles. Specifically, our approach tackles this issue on two levels: first, by representing objects using affordance and geometric part information; second, by dividing the robot's observed images into multi-scale patches. By correlating these patches with geometric parts and affordance information, the robot can more accurately infer and locate target objects from partially observed attributes. In contrast, previous methods, which recognize object-level categories typically obtained from a relatively complete view of an object, struggle to identify the object from partial observations, especially when only a small part of the object is observed by the robot.
> In addition to geometric part information, we propose leveraging affordance information for navigation. Affordances describe the possible actions that can be performed with an object, and affordances remain identifiable even from partial views, offering robustness to variations in appearance due to lighting or occlusions. Affordance for navigation has seldom been explored in previous literature. In our work, more concretely, affordance information (e.g., it can provide support) enhances the confidence in the existence of a chair in the area to be explored, thus increasing the exploration efficiency.
>
> ---
>
> We agree with your suggestion to use a table to illustrate the differences between our work and existing works. For this purpose, we have created ***Table a***, as shown in the attached PDF. The key difference between our method and other methods is that we leverage geometric part information and affordance information to represent the environment, in addition to using object-level category information as previous methods do. Furthermore, we utilize multi-scale feature representation to capture local features, enhancing the overall accuracy and robustness of our approach.
>
> By clarifying these innovations and adding the comparative table, we hope to more clearly demonstrate our work's key novelties and further validate our research contributions.
>
> ---
> # Response to Weakness 2
>
> Thank you to the reviewer for the thorough review and feedback on our work. Due to the time constraints of the rebuttal period, we have only completed initial testing in an indoor environment for the task of zero-shot object goal navigation on a real-world robot. The test results are shown in the attached PDF in ***Figure b***.  As shown in clockwise order in ***Figure b***, at the beginning, the robot is placed at a random position, and the target object is a trash can. The robot begins to scan the environment using geometric parts and affordance information. However, the robot does not find any trash can-related geometric parts or affordance attributes in the bedroom. Then, the robot moves to the living room for further exploration. Here, it continues to utilize affordance information to prioritize the exploration area that more likely contains the target object. Next, the robot moves towards the kitchen and identifies the trash can there, marked with a red circle in the image, indicating the successful recognition of the target.
>
> ---
> # Response to Weakness 3
>
> Thank you for your thorough review. We will remove the checklist instruction blocks in the revised version.
>
> ---
> # Response to Weakness 4
>
> Thank you for your detailed suggestions. We will thoroughly proofread the entire paper and address issues such as the citation problem in Line 214. Our aim is to ensure accurate citations, fluent language, and clear structure throughout the paper. We will include these revisions in the updated version of the paper to enhance its quality and credibility.
>
> ---
> # Response to Questions
>
> Thank you very much for your comprehensive feedback and suggestions on our paper. We have provided point-by-point responses to your concerns in the weakness part and will include these clarifications in the final version of the paper. These include further explanations of the core innovations, a comparison table with existing work, and real-world experimental results. Additionally, we will thoroughly proofread the entire paper to address formatting, citation, and textual issues, thereby improving the overall quality.
>
> ---
>
> We aim to address all your concerns comprehensively, demonstrating the innovation and effectiveness of our method. We have also responded to the other reviewers' comments in detail, and we kindly request your further review. We believe we have thoroughly addressed all the feedback and hope this will lead to an improved evaluation. Thank you again for your time and constructive comments on this paper.

---

> > ### Comment · Reviewer_fGDQ · 2024-08-12
> >
> > Thank the authors for the rebuttal. I am still not fully convinced by the real world experiment which I think is very important to validate the effectiveness of the proposed method. Only one experiment in one scenario without comparing to existing methods cannot convince me of that. Therefore, I am still keeping my original rating.

---

> > > ### Author Response · Authors · 2024-08-12
> > > **In response to fGDQ’s further feedback**
> > >
> > > Thank you for your further feedback and suggestions. We understand your concerns regarding the real-world experiments and the importance of validating our proposed method across more diverse scenarios and through comparisons with existing methods. In fact, since we first received your suggestion about real-world experiments, we have been working diligently to enhance our experiments. Due to the one-week rebuttal period, we only presented a demo in one real-world scenario during this stage. Before the camera-ready deadline, we will definitely include the completed real-world experiments.
> > >
> > > In our real-world experiment, we will evaluate the performance of four zero-shot object goal navigation algorithms, including L3MVN, COW, ESC, and VLFM, within a standard indoor apartment environment consisting of two bedrooms, two bathrooms, one kitchen, and one living room. The experiment involved both a Unitree B1 quadruped and a small, four-wheeled robot, each equipped with an RGB-D sensor, to navigate the environment and locate specific target objects, including a bed, toilet, plant, table, sofa, chair, trash can, TV, and remote control, without any prior knowledge of their locations. To ensure a fair comparison, the starting position of the robots was kept consistent across all trials for each algorithm. The experiment was conducted ten times, with performance measured based on success rate to determine the most effective navigation method.
> > >
> > > If you have any additional suggestions regarding the experimental setup, we would be glad to incorporate them. We will make sure all these results will be included in the camera-ready version of the paper.
> > >
> > > We hope these steps will address your concerns and provide a more comprehensive validation of our method's effectiveness, particularly in light of your feedback on the importance of robust real-world testing across multiple scenarios and comparisons with existing methods. We appreciate your thoughtful review and will take your feedback into account as we refine our research.

---

### Official Review · Reviewer_13Wq · 2024-07-12

**Soundness:** 3
**Presentation:** 3
**Contribution:** 2
**Rating:** 6
**Confidence:** 5

**Summary:**

The paper presents an approach for the task of zero-shot object goal navigation. The approach first generates a geometric and affordance description of the goal object. This description is then matched with the visual embedding of the RGB frame (at multiple scales) in the CLIP embedding space to compute a similarity score. They project this score on an orthographic 2D map. During navigation, the agent explores the frontiers/unexplored regions with the highest score. They demonstrate the efficacy of their method on HM3D and Gibson object-goal benchmark.

**Strengths:**

- The authors present excellent ablations in the paper, justifying each design decision with ablations. They show ablations for using affordance and geometric attributes, multiple scaling levels, different visual encoders, and even different methods for calculating and merging GA scores!
- The results (success-rate) on HM3D and Gibson object-goal navigation benchmark are better than the next best method (VLFM) (1.1% on ML3D, and 1.7% on Gibson).

**Weaknesses:**

- The idea of using attributes and longer descriptions for objects instead of just object label has been well studied [A, B] and I would have expected the authors to make connections to this popular line of work on effectively utilising CLIP representations.
- I also thought that the 23.7% increase in SPL, and 37.0% in SPL claim is quite misleading specially because the only difference between them and VLFM is that VLFM uses a learned policy for navigation. It has been well established that using FMM for navigation works well enough for indoor navigation

[A] Visual Classification via Description from Large Language Models, ICLR 2023.
[B] LLMs as Visual Explainers: Advancing Image Classification with Evolving Visual Descriptions, 2023
[C] Navigating to Objects in the Real World, 2022.

**Questions:**

- The authors argue that comparison to VLFM is unfair because they use a trained policy compared to FMM to achieve high SPL on HM3D. Even then, they obtain better SPL on Gibson. Can the authors comment on why a trained policy will help HM3D, but not Gibson?
- I didn't fully understand Figure 3. What does decrease in success rate mean? Is it relative or absolute decrease? Is it percentage or absolute points?

**Limitations:**

The authors don't discuss any limitation of their work in the manuscript.
- Maybe the authors should add a small section discussing potential limitations such as lack of real-robot experiments in the paper.
- assumptions about perfect odometry / localization to build a map.

Not a limitation, but a suggestion to improve the manuscript:
- I thought that the writing could be clearer. Some of the sentences were overly complicated. For example Line 42-43 - "proposed method initiates with the LLM inferring the object’s geometric part attributes and potential
affordances attributes, delving into a detailed understanding of the object’s physical structure and
functional properties". The paper will benefit from more succinct and clearer writing.

---

> ### Author Rebuttal · Authors · 2024-08-06
>
> # Response to Weakness 1
>
> Thank you to the reviewer for the valuable suggestions.
>
> While previous work has effectively utilized attributes and longer descriptions for object classification [A, B], our approach uniquely explores the role of attributes in navigation, particularly the use of affordance attributes. We demonstrate the distinct advantages of leveraging affordance in navigation, especially for dealing with partial observations obtained by a robot. Utilizing affordance attributes in exploration and navigation provides the robot with valuable hints, allowing it to prioritize exploring areas with a higher likelihood of containing the target objects with the desired affordance. We hope this explanation more clearly demonstrates the innovation and effectiveness of our method.
>
> We hope this will more clearly demonstrate the innovation and effectiveness of our method.
>
> ---
>
> # Response to Weakness 2
>
> Thank you to the reviewer for the thorough review and feedback on our work. The percentages in Table 1 of the original text were compared against methods without locomotion and semantic training, which indeed may cause misunderstandings. We will remove the percentage improvements in the revised version.
>
> To ensure a fairer comparison with VLFM, we redesigned a baseline according to reviewer KUEn's suggestion, where VLFM uses FMM for navigation instead of the trained policy. The specific results can be found in ***Table b*** of the attached PDF. It can be observed that our model shows improvements in SR and SPL on HM3D by 4.32% and 10.17%, respectively, and on Gibson by 3.50% and 14.43%, respectively, compared to this baseline. This further validates the effectiveness and advantages of our method.
>
> Additionally, we compared our method with other FMM-based methods, as shown in Table 1 of the original paper, demonstrating the effectiveness of our proposed GAMap and multi-scale geometric and affordance scoring method.
>
> ---
>
> # Response to Question 1
>
> Thank you to the reviewer for the valuable suggestions. There are two main reasons for this:
>
> 1. According to the description in section IV.D of the VLFM paper, it seems like the policy was trained only on the HM3D dataset and not on the Gibson dataset.
>
> 2. The HM3D dataset is more complex and diverse compared to the Gibson dataset. A trained policy can learn complex environmental features and navigation strategies from extensive training data and scenes. Therefore, using a trained policy shows more significant improvement on the HM3D dataset than on the Gibson dataset.
>
> We will include these detailed explanations in the revised version of the paper. Once again, thank you for your review and feedback.
>
> ---
> # Response to Question 2
>
> Thank you to the reviewer for the detailed review of our work.
> Figure 3 illustrates the impact of different numbers of geometric attributes and affordance attributes on time and performance. Specifically, we use the performance of three geometric attributes and affordance attributes as baselines, and the performance is measured using percentage (%).
>
> To more clearly convey the Effectiveness of Affordance and Geometric Attributes, we have converted Figure 3 to a table, as shown in ***Table c*** of the attached PDF. We will replace it in the revised version of the paper to ensure readers can accurately understand the experimental results.

---

> > ### Comment · Reviewer_13Wq · 2024-08-12
> > **Thanks!**
> >
> > Thanks for providing the clarifications. I am happy with your responses. Since most of my questions and weaknesses were clarifications, I will stick to my original rating.

---

### Official Review · Reviewer_KUEn · 2024-07-13

**Soundness:** 2
**Presentation:** 3
**Contribution:** 2
**Rating:** 6
**Confidence:** 4

**Summary:**

The authors propose an algorithm for the object-goal navigation/exploration problem by using a 2D navigation map containing geometric and affordance scores for the target object. Geometric and affordance features for the target object are extracted by asking an LLM to list them for the object, and these words are then used with a multi-scale CLIP setup to get the resulting "GA" scores that are projected into a 2D map and accumulated over time.

**Strengths:**

* The object navigation problem is important for robotics and embodied AI applications
* The paper is mostly well-written and the experiments have ablations
* The approach appears novel if somewhat narrow.

**Weaknesses:**

* Narrowness of contribution: Projecting CLIP-like open-vocab scores into a 2D navigation map has been done in earlier work like VLFM [17]. The novelty here appears to be that you ask an LLM for multiple (geometric, affordance) properties of the object to query CLIP with.

* Motivation of baselines: The results only seem about 1% better than the recent VLFM paper, which you dismiss because it contains "locomotion" training. However, no mention is made how your robot does motion planning. Since your results measure path length, you have to use something to get to a high-scoring frontier without colliding. The digitized environments are also just referred to as "data sets", but there has to be a simulator running on top of them to simulate some type of robot moving through the environment. Some details on this may help motivate why you dismiss VLFM.

* Reproducibility: The results are all with (and contributions hinges on) GPT4, but the back-end of GPT4 changes substantially over time and there is afaik no way to run a specific version of GPT4 (only major versions). This means the numbers in the paper may not be reproducible at the time of publication. It would be helpful to also include results with a specific version of an open-source LLM to at least have something reproducible.

--------
After rebuttal: The authors ran more experiments which addressed weaknesses #2 and #3. The quantitative improvements weren't as good as originally claimed but good enough. They even threw in a real robot experiment (video would also have been nice!)..The contribution is still a bit narrow but given the SOTA results on ObjNav it merits publication..

**Questions:**

* Please clarify what kind of path planning and robot model you use, and in more detail why VLFM is not comparable.

* Even assuming the VLFM "locomotion" is not comparable, could you not just use their 2D map component with your path planning? Both are 2D maps with scores.

**Limitations:**

Adequate except for the points above.

---

> ### Author Rebuttal · Authors · 2024-08-06
>
> ---
> # Response to Weakness 1
>
> We acknowledge the contributions of existing works, such as VLFM, in utilizing CLIP-like models for navigation. VLFM primarily considers semantic relevance between all objects in the scene and the target object to generate value maps, which is indeed effective.
> Different from VLFM we propose **Geometric and Affordance Map (GAMap)** and the **Multi-scale Geometric and Affordance Scoring** to address the challenges in semantic scene understanding during open-vocabulary robotic exploration in unknown environments, caused by occlusion or partial viewing angles. Our approach tackles this issue on two levels: first, by representing objects using affordance and geometric part information; second, by dividing the robot's observed images into multi-scale patches. By correlating these patches with geometric parts and affordance information, the robot can more accurately infer and locate target objects from partial observations. In contrast, previous methods, which recognize object-level categories typically obtained from a relatively complete view of an object, struggle to identify the object from partial observations, especially when only a small part of the object is observed by the robot.
>
> In addition to geometric part information, we propose leveraging affordance information for navigation. Affordances describe the possible actions that can be performed with an object, and affordances remain identifiable even from partial views, offering robustness to variations in appearance due to lighting or occlusions. Affordance for navigation has seldom been explored in previous literature. In our work, more concretely, affordance information (e.g., it can provide support) enhances the confidence in the existence of a chair in the area to be explored, thus increasing the exploration efficiency.
>
> ---
> # Response to Weakness 2
>
> Thank you for the detailed review and feedback. We understand your concerns. Below is our detailed response and clarification.
>
> We mentioned that VLFM includes "locomotion" training because it employs a distributed deep reinforcement learning algorithm to train a PointNav policy to help the robot navigate to a designated waypoint. The authors trained their PointNav policy using scenes from the HM3D dataset with 4 GPUs, each with 64 workers, for 2.5 billion steps, taking around 7 days.
> In our work, we utilize the heuristic search algorithm FMM for motion planning, as mentioned in Line 200 of the original paper, to find the shortest path from the robot's current location to the frontier. Additionally, we adopted the Habitat simulator but did not use any of its provided path planning algorithms. Instead, we implement FMM for path planning.
>
> We compared the accuracy of VLFM and our method using the same motion planning algorithm, as suggested in your Question 2. For detailed results, please refer to our response to Question 2.
>
> ---
> # Response to Weakness 3
>
> Thank you for highlighting concerns regarding the reproducibility of our work. To address this, we selected LLAMA and LLAMA2 as open-source LLMs and conducted navigation experiments on HM3D and Gibson datasets. The results are shown in ***Table e*** of the attached PDF. We will include this table in the final version of the paper and release all code on GitHub.
>
> ---
> # Response to Question 1
>
> Thank you for your valuable feedback. Below is our detailed response to the concerns you raised:
>
> **Path Planning**
>
> Our method uses FMM to find the shortest path from the robot's current location to the designated target point.
>
> **Robot Model**
>
> We employed a simulation environment based on the Habitat simulator and did not use provided path planning method.
>
> **Why VLFM is not comparable**
>
> As mentioned in our response to Weakness 2, VLFM uses a trained PointNav policy for robot navigation. This training process requires a large amount of training data and computational resources and is complex and time-consuming. Specifically, it involves approximately 7 days of training on 4 GPUs, each with 64 workers, using the training split of the HM3D dataset.
>
> In contrast, our method uses FMM, which does not need training, simpler to implement, and more computationally efficient.
>
> To ensure a fair comparison, we designed a comparative experiment according to the suggestion mentioned in your Question 2. For detailed results, please refer to our response to Question 2.
>
> ---
> # Response to Question 2
>
> Thank you for the constructive suggestions. We also believe this is an excellent baseline to better demonstrate the effectiveness of our method. Following your suggestion, we use the Vision-Langauge Frontier Map proposed by VLFM for frontier selection and use FMM for path planning. The experimental results are shown in ***Table b*** of the attached PDF. It can be observed that, after removing the PointNav motion planning module, the VLFM variant shows performance decline in SR and SPL on both datasets. Our model shows improvements of 4.32% and 10.17% in SR and SPL on HM3D, and 3.50% and 14.43% on Gibson, respectively, compared to this new version of VLFM.
>
> These results indicate that with the same path planning algorithm, our model still shows significant improvements in navigation performance compared to VLFM. This demonstrates that our proposed GAMap is more effective for robotic navigation tasks than the value map generated by VLFM. This further validates the innovation and superiority of our model.
>
> ---

---

> > ### Comment · Reviewer_KUEn · 2024-08-11
> > **Response to Author Rebuttal**
> >
> > I thank the authors for doing the extra experiments, they make the paper stronger. I suggest you include the VLFM + motion planning results to make it more of an apples to apples comparison rather than just saying that VLFM is not comparable.
> >
> > I will raise my score pending reviewer discussion. That said, the improvements compared to VLFM + motion planning aren't very large (3-4% SR and 10-15% SPL) but they seem big enough to merit publication. The G+A approach is somewhat novel, although the other reviewers are right to question how specific to it is to CLIP. The entire affordance part also had rather marginal improvements as seen in the ablations (going from 1 -> 3 had no effect). Based on how limited CLIP language understanding tends to be, I had also not expected this to work well. If it is a good idea, maybe it will work better with some future VLM.
> >
> > As a tangent, it is not obvious what VLFM gained by training a custom PointNav policy instead of using an off-the-shelf motion planning algorithm like the authors. The VLFM paper does not seem to be clear on that. Any standard motion planner should be able to provide a near optimal (shortest) path for a simple robot like here (I assume it is point or sphere). It is curious that the original VLFM w/ trained-PointNav actually performed better than the version with the (optimal?) motion planner by the authors. Do the authors have any intuition of why it is more suitable for the ObjNav objective (or conversely why theirs is less so)?

---

> > > ### Author Response · Authors · 2024-08-12
> > > **In response to KUEn’s further feedback**
> > >
> > > Thank you for your detailed review and thoughtful suggestions. We appreciate the time you’ve taken to provide feedback on our work, and we are pleased that the additional experiments have strengthened the paper. Below, we summarized your feedback into three main comment points and provide point-by-point responses to each.
> > >
> > > # Response to Comment 1：
> > >
> > > Thank you for the suggestion. We agree that including the VLFM + motion planning results will provide a more direct and fair comparison. We will include these results in the revised version of the paper to ensure a more "apples to apples" comparison, rather than just stating that VLFM is not comparable. This addition will help to better illustrate the differences and improvements brought by our approach.
> > >
> > > # Response to Comment 2：
> > >
> > > Thank you very much for your willingness to raise the score. We acknowledge that the percentage improvements over VLFM are not very large, but our results still push the state of the art forward, and as you mentioned, they seem big enough to merit publication.  Additionally, we appreciate your recognition of the novelty of the G+A approach. As mentioned in our response to other reviewers, we tested other models like BLIP and BLIP-2 in our ablation study, and the results indicate that applying our method to more advanced VLMs leads to further improvements in accuracy. Agreeing with your suggestions for our future work, we plan to integrate more advanced VLMs to further improve our robust navigation approach.
> > >
> > > # Response to Comment 3：
> > >
> > > Our intuition is that this training helps the PointNav policy recognize patterns, make predictions, and optimize decisions that are more aligned with the ObjNav objective. In contrast, off-the-shelf motion planning algorithm  ​​, which hasn’t been exposed to as many scenarios, might not perform as effectively. For example, in the VLFM paper, PointNav was trained on the HM3D dataset, enabling it to learn locomotion tailored to the dataset’s specific scenarios. This specialization explains its strong performance in subsequent navigation tasks within HM3D.

---

### Official Review · Reviewer_suJv · 2024-07-15

**Soundness:** 3
**Presentation:** 3
**Contribution:** 2
**Rating:** 5
**Confidence:** 3

**Summary:**

The paper proposes Geometric and Affordance Maps to tackle zero-shot object-goal navigation. It focuses on how to leverage VLMs to navigate toward objects of unfamiliar categories without prior training. The authors use GPT-4 to generate multiple attributes that describe affordance and geometry, and acquire attribute embeddings by CLIP text encoder. Besides, they propose a multi-scale scoring approach which divides each RGB image into multiple patches processed by CLIP visual encoder, to acquire multi-scale visual embeddings. Cosine similarity can be computed between those embeddings, resulting in a map used for navigation (e.g., applying Fast Marching Method based on the map). Experiments on HM3D and Gibson demonstrate improvements in success rates and success weighted by path length (SPL). The authors also provide an analysis on error pattern, which shows that detection error (missing the target object) is significantly reduced.

**Strengths:**

The paper is clearly written. The proposed approach to leverage LLMs and VLMs to tackle zero-shot object-goal navigation is clean and simple.

**Weaknesses:**

1. The overall approach is limited by LLMs and VLMs. The analysis in Sec. 5.9 (Table 4) implies that the improvement mostly results from the reduced detection error. Since off-the-shelf LLMs and VLMs are used, the technical contribution of the paper seems to be limited by this fact. Although geometric and affordance attributes are proved effective in Figure 3, the conclusion might be limited to the specific VLM used in this paper, namely CLIP.
2. It is better to replace Figure 3 with a table. It is not easy to tell the exact effect of $N_a$ and $N_g$.
3. Some sentences in Sec. 5.1 are with typos (perhaps in a rush). For example, "Gibson is generated by Al-Halah et al. [36] for agents with configuration A". And the citations in L214 are messy.

**Questions:**

1. Can the authors compare the proposed method with PIVOT[1], which can also leverage VLMs to tackle robotic navigation? PIVOT focuses more on VQA, while this paper focuses on the similarity between the text and visual embeddings generated by CLIP. CLIP may not capture enough details, and thus the authors may need to introduce a multi-scale scoring approach. It is interesting to see whether VLM models can directly capture enough multi-scale information.
2. Can the authors also compare the time cost with baselines, like SemExp and VLFM?


[1] PIVOT: Iterative Visual Prompting Elicits Actionable Knowledge for VLMs

**Limitations:**

The authors have addressed the limitations in the appendix.

---

> ### Author Rebuttal · Authors · 2024-08-06
>
> ---
> # Response to Weakness 1
> Thank you to the reviewer for the feedback on our paper. We understand your concerns and would like to clarify and respond to them as follows:
>
> 1. We would like to first highlight the main contributions of this paper: the Geometric and Affordance Map (GAMap) and the Multi-scale Geometric and Affordance Scoring Method. These innovations primarily address the challenges in semantic scene understanding during robotic open-vocabulary exploration in unknown environments, particularly those caused by occlusion or suboptimal viewing angles.
>
> 2. In the analysis of Section 5.9 (Table 4), we indeed show significant improvements in reducing detection errors with our approach. These improvements are primarily attributed to the aforementioned GAMap and the multi-scale geometric and affordance scoring method. Specifically, as shown in Table 1 of the original paper, our method (CLIP+GAMap) significantly outperforms CoW (CLIP) with an increase in performance from 32 to 53.1. Additionally, our method (CLIP+GAMap) outperforms VLFM (BLIP-2), showing a performance improvement of 3.2 in SR. Furthermore, as demonstrated in the ablation studies in Table 2 of the original paper, the performance of BLIP-2+GAMap is better than that of BLIP-2.
> Therefore, we can draw the conclusion that the improvements in reducing detection errors are primarily attributed to the aforementioned GAMap and multi-scale geometric and affordance scoring method. Also, our GAMap and and multi-scale geometric and affordance scoring can be broadly applicable and not limited to a specific VLM, as evidenced by the performance improvement on different VLMs, such as CLIP, BLIP, and BLIP-2.
>
> We sincerely appreciate the valuable feedback from the reviewer. We believe that through these clarifications and explanations, we can better showcase the technical contributions and innovations of our work.
>
> ---
> # Response to Weakness 2
>
> Thank you for your suggestion. We agree that converting Figure 3 into a table can more clearly demonstrate the specific contributions of Na and Ng to various performance metrics. We have redrawn the table as shown in ***Table b*** of the attached PDF. We will include this table in the revised version of the paper to better present our experimental results.
>
> ---
> # Response to Weakness 3
>
> We appreciate the reviewer's careful examination and understanding of our paper. We have revised Sec 5.1, correcting all spelling errors and citation format issues.
>
> ---
> # Response to Question 1
>
> Thank you to the reviewer for the valuable suggestions. Since PIVOT has not released its code, we have chosen to compare using the VLM employed in their paper, namely GPT4v. Given the difficulty of conducting a comprehensive quantitative comparison within a week, we have attempted to address whether VLM models can directly capture enough multi-scale geoemtric and affordance information through a qualitative example.
>
> The experiment was designed as follows: we randomly selected a scene and compared the ability of GPT4v and our proposed CLIP with multi-scale scoring method to identify the target object. As shown in ***Figure a*** of the attached PDF, we input an image with a sofa located in a distant corner as the target object and compared the subsequent movement trajectories of the two algorithms.
> As illustrated in the figure, our method successfully captures the small sofa back in the far corner, leveraging geometric part and affordance attributes to guide the exploration process. In contrast, GPT-4V failed to identify the object.
>
> ---
> # Response to Question 2
>
> Thank you for the reviewer’s suggestion. We selected three baselines, namely SemExp, L3MVN, and VLFM, and compared their FPS and performance metrics on navigation tasks in the HP3D dataset. The experimental results are shown in ***Table d*** of the attached PDF.
>
> It can be observed that SemExp has the highest FPS, indicating the fastest processing speed. This is because it uses a detection head and does not employ a Foundation model. However, SemExp has the lowest SR and SPL, indicating that despite its fast processing speed, it performs poorly in navigation accuracy and path efficiency. In contrast, L3MVN has the second-highest FPS as it uses a lightweight foundation model. Although its processing speed is not as fast as SemExp, it shows improvements in navigation accuracy and path efficiency, achieving an SR of 76.1 and an SPL of 37.7. On the other hand, VLFM has a lower FPS of only 2, but it significantly improves SR and SPL, reaching 84.0 and 52.2, respectively. This indicates that although VLFM has a slower processing speed, it has considerable advantages in navigation accuracy and path efficiency. Our model has the same FPS as VLFM, both at 2, but further improves SR and SPL, reaching 85.7 and 55.5, respectively. This demonstrates that our method maintains high navigation accuracy and path efficiency while providing comparable processing speed to VLFM.
>
> These experimental results verify that our proposed method achieves a good balance between time and accuracy. This indicates the excellent overall performance and practical application of our method. We will include these detailed experimental results and comparative analyses in the revised version of the paper.

---

> > ### Comment · Reviewer_suJv · 2024-08-13
> >
> > Thank the authors for the response. My concerns are resolved. I will keep my rating or raise the score based on discussion with other reviewers and AC.

---

### Author Rebuttal · Authors · 2024-08-06

# General Response

We appreciate all reviewers for their thorough reviews and valuable suggestions on our paper. First, we address three common concerns raised by all the reviewers here, and then we provide point-by-point responses to each reviewer's specific comments.

---

## 1. Novelty of the Paper
The main contributions of this paper are twofold: the **Geometric and Affordance Map (GAMap)** and the **Multi-scale Geometric and Affordance Scoring**. These contributions primarily address the challenges in semantic scene understanding during open-vocabulary robotic exploration in unknown environments, caused by occlusion or partial viewing angles. Our approach tackles this issue on two levels: first, by representing objects using affordance and geometric part information; second, by dividing the robot's observed images into multi-scale patches. By correlating these patches with geometric parts and affordance information, the robot can more accurately infer and locate target objects from partial observations.

For instance, when a robot observes only a patch of a chair back, our method can infer the geometric part information (e.g., this is a chair back), thus identifying the object as a chair. In contrast, previous methods, which recognize object-level categories typically obtained from a relatively complete view of an object, struggle to identify the object from partial observations, especially when only a small part of the object is observed by the robot.

In addition to geometric part information (e.g., this is a chair back), we propose leveraging affordance information (e.g., it can provide support) for navigation. Affordances describe the possible actions that can be performed with an object, and affordances remain identifiable even from partial views, offering robustness to variations in appearance due to lighting or occlusions. Affordance for navigation has seldom been explored in previous literature. In our work, more concretely, affordance information (e.g., it can provide support) enhances the confidence in the existence of a chair in the area to be explored, thus increasing the exploration efficiency.

Extensive experiments and ablation studies demonstrate the effectiveness of the aforementioned contributions, as shown in Table 1 to Table 4 and Figure 3 to Figure 5 of the original paper. Through these clarifications and explanations, we hope that our technical contributions and innovations are more clearly presented.

## 2.Comparison with VLFM
VLFM first generates an object-level language-grounded value map based on the semantic relevance of all the objects in the environment w.r.t the category of the target goal, and then uses this map to identify the most promising frontier to explore. In contrast, our approach leverages multi-scale geometric part and affordance information to construct a Geometric and Affordance Map (GAMap) for enhanced semantic scene understanding to guide robot exploration. In particular, our GAMap reveals the geometric part and affordance relevance of each object in the environment w.r.t the geometric part and affordance attributes of the target goal.  The differences between VLFM and our method make our method more robust in identifying partially visible objects.

Another difference lies in the path planning method. After obtaining the 2D value map, VLFM employs a distributed deep reinforcement learning algorithm, i.e., Variable Experience Rollout (VER), to train a Point Goal Navigation (PointNav) policy to help the robot navigate to a designated waypoint. The authors trained their PointNav policy using scenes from the training split of the HM3D dataset with 4 GPUs, each having 64 workers, for 2.5 billion steps, which took around 7 days. In contrast, after generating the GAMap, our method employs a heuristic search algorithm, i.e., Fast Marching Method (FMM), to find the shortest path from the robot’s current location to the designated waypoint. ***Unlike PointNav, which requires about a week of pre-training, our FMM does not require any training process.*** This makes our approach easier to apply to various environments with different conditions, as the trained Point Goal Navigation might not easily transfer to unseen environments. Additionally, FMM is simpler to implement and computationally more efficient.

Considering the different path planning methods adopted in our approach and VLFM, and based on **Reviewer KUEn**'s suggestion, we redesigned a comparative experiment with VLFM. Specifically, we kept the VLFM value map generation process unchanged and replaced its path planning method with FMM instead of the trained policy. The detailed results are shown in ***Table b*** of the attached PDF. The comparison reveals that our model outperforms this baseline by 4.32% and 10.17% in SR and SPL on HM3D, and by 3.50% and 14.43% in SR and SPL on Gibson, respectively. We hope this comparative experiment further validates the effectiveness and advantages of our GAMap compared to the vision-language frontier map proposed in VLFM.

## 3. Modification of Figure 3

To more clearly demonstrate the specific impacts of Na and Ng on various performance metrics, we have converted Figure 3 to a table, as shown in ***Table c*** of the attached PDF. We will include this table in the revised version of the paper.

---
Through the above explanations and experiments, we hope to address the major concerns raised by the reviewers and further demonstrate our contributions. We will continue to provide detailed responses to each reviewer's specific comments and reflect these improvements in the revised version. Once again, we thank all reviewers for their valuable feedback and suggestions.

---

### Decision · Program_Chairs · 2024-09-25

**Decision:**

Accept (poster)

**Comment:**

This paper received divergent initial opinions with two reviewers in support of acceptance and two negative reviewers.  The main concerns by the negative reviewers revolved around narrowness and reproducibility of the contribution, missing time cost comparisons for the method and baselines, missing comparisons to prior work using VLMs for navigation, and missing experiments with real-world robot navigation.

The rebuttal responded to reviewer concerns, and one initially negative reviewer raised their opinion to accept.  Another negative reviewer acknowledged that some of their concerns were alleviated but remained opposed to acceptance citing the lack of convincing real-world robot experiments as the rationale.  That concern was not shared by any of the other reviewers, and in fact the authors do describe a real-world experiment in their rebuttal (albeit at a small scale).  Thus, the AC does not find a basis to overrule the reviewer consensus in favor of acceptance.  The AC recommends acceptance and strongly encourages the authors to incorporate clarifications and revisions to improve the final manuscript.